# Reduced Abundance of Nitrate-Reducing Bacteria in the Oral Microbiota of Women with Future Preeclampsia

**DOI:** 10.3390/nu14061139

**Published:** 2022-03-08

**Authors:** Faisal Altemani, Helen L. Barrett, Leonie K. Callaway, H. David McIntyre, Marloes Dekker Nitert

**Affiliations:** 1School of Chemistry and Molecular Biosciences, The University of Queensland, Brisbane, QLD 4072, Australia; faisal.altemani@uq.net.au; 2Department of Medical Laboratory Technology, Faculty of Applied Medical Sciences, University of Tabuk, Tabuk 71491, Saudi Arabia; 3Mater Research, The University of Queensland, Brisbane, QLD 4001, Australia; helen.barrett@mater.uq.edu.au (H.L.B.); h.d.mcintyre@uq.edu.au (H.D.M.); 4Department of Obstetric Medicine, Royal Brisbane and Women’s Hospital, Brisbane, QLD 4006, Australia; leonie.callaway@health.qld.gov.au; 5Centre for Clinical Research, The University of Queensland, Brisbane, QLD 4006, Australia

**Keywords:** preeclampsia, blood pressure, oral microbiota, nitrate, *Veillonella*, leafy green vegetables

## Abstract

The oral microbiota can contribute to the regulation of blood pressure by increasing the availability of nitric oxide through the reduction of nitrate to nitrite, which can be converted into nitric oxide in the stomach and then enter the circulation. It is unclear if the composition of the oral microbiota is different between women who do and do not develop preeclampsia. This study aimed to compare the composition of the buccal microbiota just prior to the development of symptoms at 36 weeks gestation in 12 women who developed late-onset preeclampsia and 24 matched women who remained normotensive throughout pregnancy by 16S rRNA gene amplicon sequencing. The abundance of the nitrate-reducing *Veillonella* spp *V. parvula* and *V. dispar* and a subunit of nitrate reductase narH was compared using real-time PCR. The abundance of bacteria was correlated with maternal blood pressure and dietary intake of nitrate-containing vegetables. The results showed that the abundance of nitrate-reducing bacteria including *Veillonella,* specifically *V. parvula*, and *Prevotella* was reduced in women who developed preeclampsia. *Veillonella* but not *Prevotella* abundance was negatively correlated with maternal blood pressure. The dietary intake of nitrate-containing vegetables did not differ between the groups and was not correlated with the abundance of *Veillonella*. There was no difference in the abundance of the nitrate reductase subunit narH between the groups. These results suggest that the abundance of nitrate-reducing bacteria is reduced in the oral microbiota of women who later develop preeclampsia, indicating a potential pathway for prevention.

## 1. Introduction

The human oral microbiota is a complex microbiome that varies in its composition between different oral locations such as the tongue, the palate, the tooth, the gingiva, the saliva and the buccal epithelium [1]. The composition of the microbiota in the saliva, hard palate and buccal mucosa is more similar than that of the gingiva, the tooth and dental plaque [1]. However, certain genera are common in all locations including *Veillonella*, *Prevotella*, *Haemophilus*, *Rothia*, and *Actinomyces* [1]. The oral microbiota often forms thin biofilms where bacteria live and act in a synergistic manner and where an aerobic gradient is formed [2].

The oral microbiota has recently been identified as a contributor to blood pressure regulation through its indirect production of the vasodilator nitric oxide (NO) [3,4]. Urinary NO concentrations increase in pregnancy [5], as do serum NO concentrations [6]. Pregnancy complications such as preeclampsia and fetal growth restriction have been associated with lower maternal urinary or circulating nitrate and nitrite concentrations [5,7].

Dietary nitrates from leafy vegetables can be converted in the oral cavity by commensal bacteria (including the genera *Veillonella*, *Actinomyces* and *Prevotella* [8]) to nitrite with the help of nitrate reductase, which is then converted to NO in the acidic environment of the stomach [9]. The (facultative) anaerobic bacteria use nitrate as an alternative electron in the absence of sufficient oxygen to continue ATP production, producing nitrite [2]. The nitrite is then transported to the acidic stomach, where it is reduced to NO, which is absorbed and enters the circulation and is secreted into the saliva (a process known as the enterosalivary nitrate circulation) or remains in the circulation [2]. Bacteria can reduce nitrate via three classes of reductases: the membrane-bound respiratory reductases (encoded by the Nar genes), the cytoplasmic assimilatory reductases (encoded by the Nas genes) and the periplasmic dissimilatory reductases (encoded by the Nap genes) [10]. *Veillonella* spp., followed by *Actinomyces* spp., have been identified as high nitrate reducers from the dorsal tongue although nitrate-reducing bacteria were not detected in the oral microbiomes of all healthy individuals [11].

In a study of 23 hypertensive adults outside pregnancy, the salivary NO concentration was decreased to one-third that of 25 normotensive individuals [12]. Salivary NO was correlated with leafy greens consumption and the abundance of *Actinomyces concomitans* as measured by real-time PCR in supragingival plaque [12]. Similarly, in normotensive individuals, a higher presence of nitrate-producing bacteria in subgingival plaque was associated with lower systolic blood pressure [13].

The oral microbiome in pregnancy has been reported to be altered from the non-pregnant state but remains stable in its overall diversity and composition during pregnancy [14]. Studies using culturing techniques on gingival plaques in women with preeclampsia have shown increases in the rates of periodontitis and it associated pathogens, *Porphyromonas gingivalis*, *Tenerella forsynthensis* and *Eikenella corrodens* [15,16]. However, there have been no studies using current sequencing techniques to evaluate the composition of the oral microbiome in women who develop preeclampsia. In this study, we aimed to compare the composition of the oral microbiota in late pregnancy between women who subsequently developed preeclampsia and those who remained normotensive. The hypothesis was that women who developed preeclampsia have a lower abundance of nitrate-reducing bacteria in their oral cavity.

## 2. Methods

### 2.1. Study Population

The participants were selected from the participants of the Study of Probiotics IN Gestational diabetes (SPRING) [17]. SPRING aimed to prevent gestational diabetes mellitus in overweight and obese women by probiotic supplementation. For this sub-study, women who developed preeclampsia and had provided oral swabs at 36 weeks gestation were matched to normotensive women for treatment arm, BMI, maternal age, parity and ethnicity. Preeclampsia was defined according to the diagnostic criteria of the Society of Obstetric Medicine of Australia and New Zealand (SOMANZ) [18]. Four of the women who developed preeclampsia during pregnancy used anti-hypertensives and two of these had chronic hypertension. None of the participants in the sub-study had used antibiotics in the past eight weeks. At 28 weeks gestation, women provided dietary intake information by completing the Victorian Cancer Council Food Frequency Questionnaire V2.0 [19] and provided a fasting blood sample for measurement of metabolic parameters. The intake of all vegetables and vegetables high in nitrates (spinach, lettuce, celery, beetroot, cabbage, onion and garlic) was calculated in g/day. At 36 weeks gestation, blood pressure was measured twice with the participants seated and the mean of the measurements was used. A rayon swab (Copan 155CIS, Interpath, Heidelberg, VIC, Australia) was used to obtain a microbiome sample from the buccal surface, and the swab was air-dried before storage at −80 °C.

### 2.2. Oral DNA Extraction

Oral DNA was isolated from maternal buccal swabs obtained at 36 weeks gestation by placing the swab in a 2 mL screw-cap tube containing 0.4 g of sterile zirconia beads (0.1 and 0.5 mm diameter) and 300 μL lysis buffer (NaCl 0.5 mol/L, Tris–HCl 50 mmol/L, pH 8.0, EDTA 50 mmol/L and SDS 4% *w*/*v*) and homogenised for 3 min before incubation at 70 °C for 15 min. DNA was isolated from the oral lysates using the Maxwell 16 Buccal Swab LEV DNA Purification kit (Promega, Madison, WI, USA), following the manufacturer’s recommendations. Extracted DNA was quantified by Nanodrop ND 1000 spectrophotometer (Nanodrop Technologies, Wilmington, DE, USA). Contamination was monitored through assessment of reagent controls without addition of DNA.

### 2.3. 16S rRNA Gene Amplicon Library Preparation and Sequencing

The V6–V8 variable region of the bacterial 16S rRNA gene was PCR amplified using the 926F and 1392R primers including overhang adapters compatible with the Nextera Index PCR XT kit (Illumina Corp., San Diego, CA, USA) to produce specific bar-coded amplicons for individual samples. DNA isolated from *Escherichia coli* JM109 was used as the positive control and sterile water as the negative control. The PCR product was cleaned with AMPure XP beads (Beckman Coulter Ltd., Lane Cove, NSW, Australia). A Nextera XT Index kit (Illumina, San Diego, CA, USA) was used for sequencing the library preparation following the manufacturer’s instructions. The pooled products were sequenced at the Australian Center for Ecogenomics at the University of Queensland, Australia using an Illumina MiSeq system.

### 2.4. Bioinformatics Analysis for 16S rRNA

#### 2.4.1. Quality Control (QC)

The raw 16S rRNA reads were investigated with FastQC (Version 0.11.5) [20] and MultiQC (Version 1.7) [21] to calculate and visualize sequence quality metrics. Adaptor sequences were removed with Cutadapt (Version 1.15) [22] and short reads (<250 bases) and reads with insufficient base qualities were removed with Trimmomatic (Version 0.36) [23] using sliding window trimming and cutting once the average quality of four bases within the window fell below the threshold of 15 using Q-score criteria.

#### 2.4.2. Reads Processing and Taxonomy Assignment

The forward high-quality filtered reads were processed following QC using the Quantitative Insights Into Microbial Ecology software (QIIME2, version 2019.10) [24]. The sequence reads were denoised, chimeras detected and removed, and amplicon sequence variant (ASV) tables generated using the DADA2 (p-trunc-len = 0) algorithm. The feature table and relative frequencies feature table of each representative sequence were calculated and generated.

The taxonomic assignment of each representative feature sequence was determined and assigned through BLAST+ local alignment to the SILVA reference reads database (version 138) [25] and clustered at 99% identity. Features that comprised ≤ 0.01% of the total sample sequence count, and samples < 1000 reads were removed from the features table.

#### 2.4.3. QPCR Analysis of Species Abundance and Functional Capacity

To investigate the abundance of specific bacterial species (*Veillona dispar* and *Veillonella parvula*) and of one gene in the membrane-bound respiratory nitrate reduction pathway (narH), Q-PCR analysis was performed using the 16S gene (nt 27–1492) as the endogenous control. Each gene was interrogated in triplicate using 10 ng of DNA, 300 nM forward and reverse primer (see Appendix A) and 2× SYBR green mastermix (Bio-Rad Laboratories, Gladesville, NSW, Australia) using a Viia7 QPCR system (Applied Biosystems, Waltham, MA, USA). The PCR protocol included an activation step at 95 °C for 5 min, followed by 40 cycles of 95 °C for 15 s, 57 °C for 30 s and 72 °C for 45 s and melt curve analysis. The data were analysed with the ΔΔC_T_ method.

#### 2.4.4. Statistical Analysis

The clinical and microbiome data were not normally distributed and therefore non-parametric testing was used throughout the analysis. Microbiome data were normalised using total sum normalization (TSS) and transformation to square root. The data analysis of the 16S rRNA gene amplicon sequencing were performed with Calypso software [26] and GraphPad Prism version 8 (San Diego, CA, USA). Data are presented as the median with interquartile range in all instances. Two-sample comparisons between the clinical characteristics of the groups were performed with the Mann–Whitney U test and correlation analyses were performed using Spearman’s rank correlation coefficient tests. Sensitivity analyses were performed excluding the two women with chronic hypertension and separately for the three women who developed GDM but this did not alter the results. Microbiome diversity analyses for 16S rRNA data were performed using QIIME2. Alpha diversity, which is a measure of species richness or diversity within a sample, was determined by the Shannon and Chao1 indices. The Shannon index incorporates both species richness (e.g., how many species are present) and evenness (e.g., the level of abundance) of the species present. Chao1 only estimates species richness from the data abundance. Beta-diversity, which measures the diversity of the microbial communities among samples, was examined by Principal Coordinate Analysis (PCoA) plots using Bray–Curtis distance matrices and multivariate methods such as Redundancy Analysis (RDA) to explore associations among the composition of community groups. The Anosim statistic test was used to interrogate the dissimilarity matrices to examine whether the microbiota compositions were significantly dissimilar between groups. Differences in taxonomic abundance were assessed using the linear discriminant analysis of effect size (LEfSe) algorithm between groups, with a False Discovery Rate (FDR) < 0.05 for the Kruskal–Wallis test and a logarithmic Linear Discriminant Analysis (LDA) score equal to 2.0 as thresholds for significance. Network analysis was used to identify co-occurring bacteria and mutually exclusive bacteria classification among groups. Taxa abundances are associated with the group using Spearman’s rho (>0.5) and the Bray–Curtis dissimilarity distance method computed by 1000-fold permutation. Significant associations are reported as FDR < 0.25 and nodes (taxa) are colored based on their association with the group.

## 3. Results

### 3.1. Clinical Characteristic of Participants

The clinical characteristics of the 12 women who developed preeclampsia and 24 matched normotensive controls are described in Table 1. There were no significant differences between the groups except for higher systolic and diastolic blood pressure, circulating fasting insulin and C-peptide levels in women who developed preeclampsia. There were no differences in delivery or neonatal outcomes between the groups. Dietary intake of the participants was not different between the two groups apart from significantly higher cholesterol intake in women who developed preeclampsia (331 mg/day) compared with normotensive women (222 mg/day; *p* = 0.04) (Appendix A).

### 3.2. Oral Microbiota Diversity

The sequencing produced 2,596,398 sequences from 36 samples before processing by quality control and QIIME2 pipeline. After clustering, removal of chimeras and filtering, 2,466,945 sequences from 36 samples with 6326 features (operational taxonomic units (OTUs)) were retained. The median frequency per sample was 14,575 with and interquartile range (IQR) of 3366–22,399. The alpha and beta diversity metrics were calculated on a rarefied frequency-feature table with a minimum number of 5000 sequences per sample.

The within-subject alpha diversity of the oral microbiota was determined by the Shannon and Chao-1 indices. The oral microbiota of women who developed preeclampsia trended to have lower diversity though not significantly so (Shannon index, *p* = 0.09; Chao-1 index, *p* = 0.18) (Figure 1A,B).

Microbial diversity between oral samples (beta-diversity) from pregnant women who developed preeclampsia and normal controls was analysed by unsupervised Principal Coordinates Analysis (PCoA) and supervised Redundancy Analysis (RDA); its significance was tested using the Anosim statistic. The PCoA analysis for oral samples showed some clustering of the samples from the women who developed preeclampsia (Figure 1C), which was also observed in the supervised RDA clustering analysis (*p* = 0.04, Figure 1D). The variance between the samples as assessed by Anosim analysis was significantly higher in the women who developed preeclampsia (*p* = 0.045, Figure 1E).

### 3.3. Differences in Oral Microbiota Composition between Normotensive Women and Those Who Developed Preeclampsia

Differences in genus abundance in the oral microbiota of pregnant women who developed preeclampsia and normotensive controls were investigated using linear discriminant analysis of effect size (LEfSe) analyses. Eight genera were more abundant in the women who developed preeclampsia and six in the normotensive women (Figure 2). The women who developed preeclampsia had higher abundances of *Methanosaeta*, *Desulfomicrobium*, *Enterococcus*, *Mycobacterium*, *Thiobacillus*, and *Ochrobactrum*. Normotensive pregnant women had higher abundances of *Roseomonas*, *Johnsonella*, *Caulobacter* and the nitrate-reducing genera *Prevotella* and *Veillonella.*

Network analysis confirmed these differences with strong associations of *Prevotella* and *Veillonella* to the group of women who remained normotensive (Figure 3). Additionally, the abundance of another nitrate-reducing genus, *Rothia*, also showed an association with the normotensive women although not as strongly. In contrast, no genus was as strongly associated with the occurrence of future preeclampsia. The abundances of *Veillonella* and *Prevotella* were positively correlated (rho = 0.42, *p* = 0.02).

### 3.4. Correlations between Maternal Blood Pressure and Bacterial Abundance

Maternal systolic and diastolic blood pressure at 36 weeks gestation were negatively correlated with the abundance of the nitrate-reducing *Veillonella* genus whereas maternal blood pressure was positively correlated with the abundance of *Mycoplasma* and *Methanobrevibacter* (Table 2). There was no significant correlation between maternal blood pressure and the abundance of the nitrate-reducing bacterium *Prevotella*.

### 3.5. Veillonella Species Abundance and Correlation with Blood Pressure

Based on the negative correlation between maternal blood pressure and the abundance of *Veillonella,* the abundance of two major species, *V. dispar* and *V. parvula*, was determined using real-time PCR. Women who developed preeclampsia had a lower abundance of *V. parvula* in their oral microbiome (*p* = 0.04, Figure 4A), whereas the abundance of *V. dispar* was not significantly lower (*p* = 0.22, Figure 4A). There was a positive correlation between the abundances of the *Veillonella* genus from the 16S rRNA gene amplicon sequencing analysis with the *V. dispar* (rho = 0.68, *p* < 0.0001) and the *V. parvula* (rho = 0.62, *p* < 0.0005) abundances as measured by Q-PCR. There was also a strong positive correlation between the abundances of *V. dispar* and *V. parvula* in the QPCR analysis (rho = 0.85, *p* < 0.0001).

The abundance of a gene involved in the reduction of nitrate, narH, which encodes for the electron transfer subunit within the membrane-bound nitrate reductase enzyme complex, was also interrogated by real-time PCR. There was no difference in the abundance of this gene between women who developed preeclampsia and those who remained normotensive (*p* = 0.38, Figure 4B). The abundance of *V. parvula* correlated negatively with maternal diastolic blood pressure (rho = −0.47, *p* = 0.02) and trended toward a significant correlation with systolic blood pressure (rho = −0.39, *p* = 0.06) (Figure 4C,D). *V. dispar* abundance did not correlate with systolic blood pressure (rho = −0.27, *p* = 0.18) but almost with diastolic blood pressure (rho = −0.37, *p* = 0.06) whereas the abundance of narH was not correlated with systolic (rho = 0.10, *p* = 0.62) or diastolic blood pressure (rho = −0.08, *p* = 0.71). There was a significant positive correlation between the abundance of narH and *V. dispar* (rho = 0.39, *p* = 0.04) but not between narH and *V. parvula* (rho = 0.12, *p* = 0.54) or *Prevotella* (rho = −0.02, *p* = 0.90).

Lastly, the intake of vegetables and of vegetables high in nitrate was compared between the groups. There was no difference in overall vegetable intake: women who developed preeclampsia consumed 101.6 (53.8–137.5) g of vegetable/day vs. normotensive women 109.0 (93.7–145.0) g/day, *p* = 0.39. Similarly, there was no significant difference between the groups in high-nitrate vegetables: women who developed preeclampsia consumed 26.2 (19.1–34.3) g of high-nitrate vegetables/day vs. normotensive women 28.0 (15.3–36.2) g/day, *p* = 0.70. There were no significant correlations between either overall or high-nitrate vegetable intake and abundance of *Veillonella* or blood pressure (data not shown).

## 4. Discussion

The results of this study show that women who develop preeclampsia in late pregnancy have an altered composition of the buccal microbiota, with decreased abundance of the nitrate-reducing bacteria *Veillonella*, particularly *V. parvula,* and *Prevotella* in their buccal microbiota. *V. parvula* abundance correlated negatively with maternal blood pressure but not with narH abundance, indicating that it may be the abundance of other subunits or other enzymes or indeed altered activity through the pathway that is linking *V. parvula* abundance and lower blood pressure.

Different parts of the oral cavity have varying capacity for reducing nitrate, with the buccal surface having lower capacity than for instance the tongue even though there is also wide inter-individual variability [10,27]. The anaerobic *Veillonella* species in the oral cavity are often present on the tooth surface and the tongue, with *V. dispar* being more commonly detected in saliva and *V. parvula* in dental plaque [27]. In our study, *V. dispar* and *V. parvula* were detected in all pregnant women, whereas in a Japanese study of 24 older individuals of whom half were hypertensive, only *V. dispar* was detected in the tongue microbiota of all participants and *V. atypica* (42%) and *V. rogosae* (38%) were not ubiquitously present [28]. In another study of 11 healthy non-pregnant English adults, *V. atypica* was most common (91%), followed by *V. dispar* (81%) and *V. rogosae* (73%) in the dorsal tongue microbiome [29]. The differences in prevalence may reflect a difference in sample origin and/or may reflect a difference in age or physiological state.

Apart from converting dietary nitrate into nitrite, which can be further metabolised to nitric oxide and thus contribute to reduced blood pressure, *Veillonella* spp. can convert lactic acid into weaker acids which include acetate and propionate, thereby lowering the risk of cavity formation [2]. In a study of 25 healthy, young, non-pregnant individuals, those with a higher than the median abundance of nitrate-reducing bacteria on the tongue had a tendency toward lower systolic and diastolic blood pressure by 3 mmHg though this difference was not statistically significant [8].

Women who remained normotensive in pregnancy had a higher abundance of *Prevotella* in this study. While *Prevotella* can reduce nitrate to nitrite, they are also known to contribute to the metabolism of nitrogenous molecules into short-chain fatty acids, ammonia and sulfur compounds, especially in the subgingival space [2]. Given that in this study, the sample was taken from the buccal epithelium, it is possible that the overall proportion of *Prevotella* is lower than it is closer to the gingiva. In this cohort, *Prevotella* abundance was not correlated with blood pressure or narH abundance, suggesting that buccal *Prevotella* is not a major contributor to nitrate reduction.

In a small randomised controlled trial, pregnant women with chronic hypertension who received eight days of nitrate supplementation in beetroot juice had increased salivary and plasma nitrate and nitrite concentrations but no difference in blood pressure in the full cohort although some women showed lower blood pressures after the supplementation [30]. The variability in blood pressure response may reflect the abundance of oral nitrate-reducing bacteria, which was not investigated in this study. Furthermore, variations in maternal BMI or gestation at intervention may also have contributed to the differences in blood pressure response to beetroot juice supplementation. In contrast, a randomised controlled trial of four weeks of oral nitrate supplementation (6 mmol/day) reported a reduction in systolic (by 7.7 mmHg) and diastolic (by 2.4 mmHg) blood pressure in a cohort of non-pregnant individuals with hypertension independent of whether they were treated with antihypertensives [31], indicating that a longer period of supplementation may be necessary to see an effect on blood pressure. That fits with the finding that higher habitual green leafy vegetable intake, which are rich in nitrates, is associated with a lower risk of preeclampsia in a large population study from India [32]. In our study, there was no difference in the intake of high-nitrate vegetable intake between the groups; however, the resolution and/or accuracy of a food frequency questionnaire may be insufficient to give a reliable insight into leafy vegetable intake.

This study has several limitations. The sample size is relatively small and included women who were assigned probiotics. However, there were no significant differences in bacterial abundances between the women receiving probiotics or placebo capsules, which were designed to withstand the acidic content of the stomach and therefore unlikely to directly affect the composition of the oral microbiota. Furthermore, the samples were obtained before the participants were diagnosed with overt preeclampsia. It is possible that the differences between the buccal microbiota become larger in frank preeclampsia. However, it is interesting that the differences are present before the development of symptoms. Future studies need to establish, in a larger and longitudinal cohort, whether or not the changes in nitrate-reducing bacteria are present from early pregnancy and whether it could be used as a biomarker for women at high risk of developing late-onset preeclampsia. In addition, samples of other parts of the oral cavity should also be investigated to identify if the changes in nitrate-reducing bacteria are specific for the buccal microbiota or whether these are universal. Due to limitations in DNA availability, only narH abundance was investigated as an indicator of the nitrate-reducing capacity. The abundance of other enzyme subunits (narB, narG, narI) or other enzymes in the pathway (e.g., nitrite reductase (Nir)) should be investigated to provide more insight into the mechanism by which nitrate is reduced in the oral microbiota in pregnant women. It would also be of interest to compare the concentration of nitrate, nitrite and NO in the oral cavity of pregnant women with and without preeclampsia to ascertain if the differences in bacterial abundance result in differences in actual metabolite concentrations.

## 5. Conclusions

In summary, this study indicates that the abundances of nitrate-reducing bacteria in the oral microbiota are reduced in pregnant women with future late-onset preeclampsia, suggesting a potential pathway for prevention.

## Figures and Tables

**Figure 1 nutrients-14-01139-f001:**
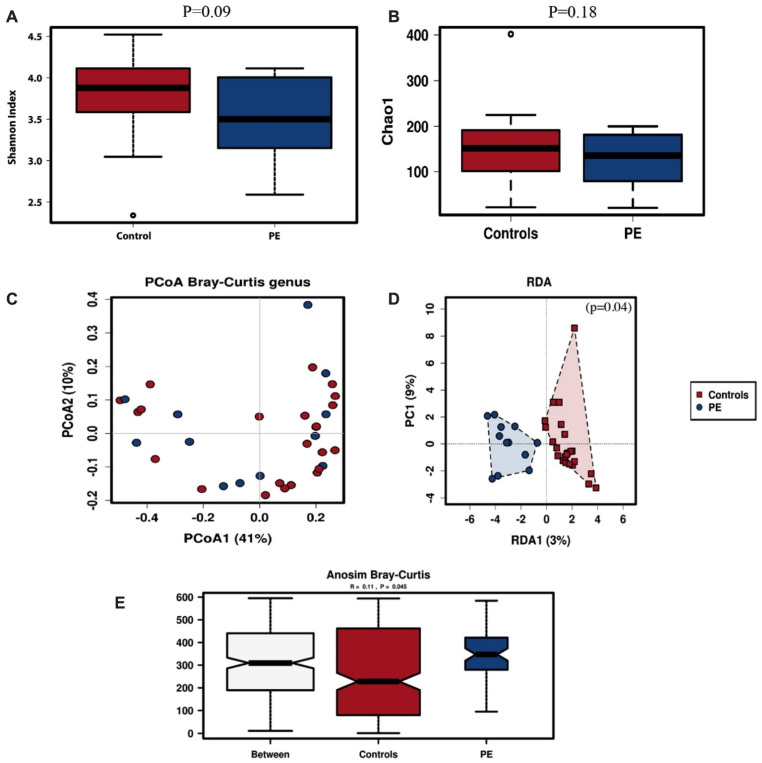
Oral microbiota diversity. Alpha diversity for oral samples of pregnant women with preeclampsia (DPE) (*n* = 12) compared to control (*n* = 24) group at 36 weeks pregnancy using Shannon (**A**) and Chao-1 (**B**) indices. Beta diversity was assessed with PCoA (**C**), RDA (**D**) and Anosim (**E**) analysis. Red symbols signify the women in the control group and blue symbols the women who developed preeclampsia.

**Figure 2 nutrients-14-01139-f002:**
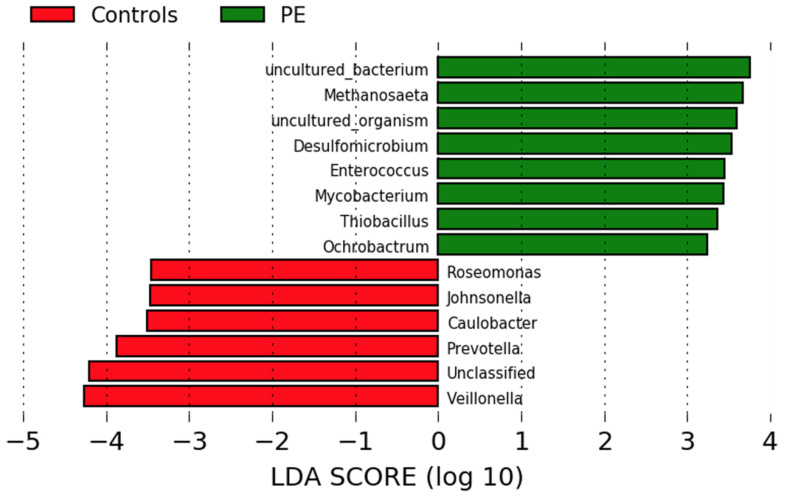
Comparison of the gut microbiota composition in women who developed preeclampsia and normotensive women. LEfSe analysis at the genus level of the oral microbiota of women who developed preeclampsia (green) and who remained normotensive (red).

**Figure 3 nutrients-14-01139-f003:**
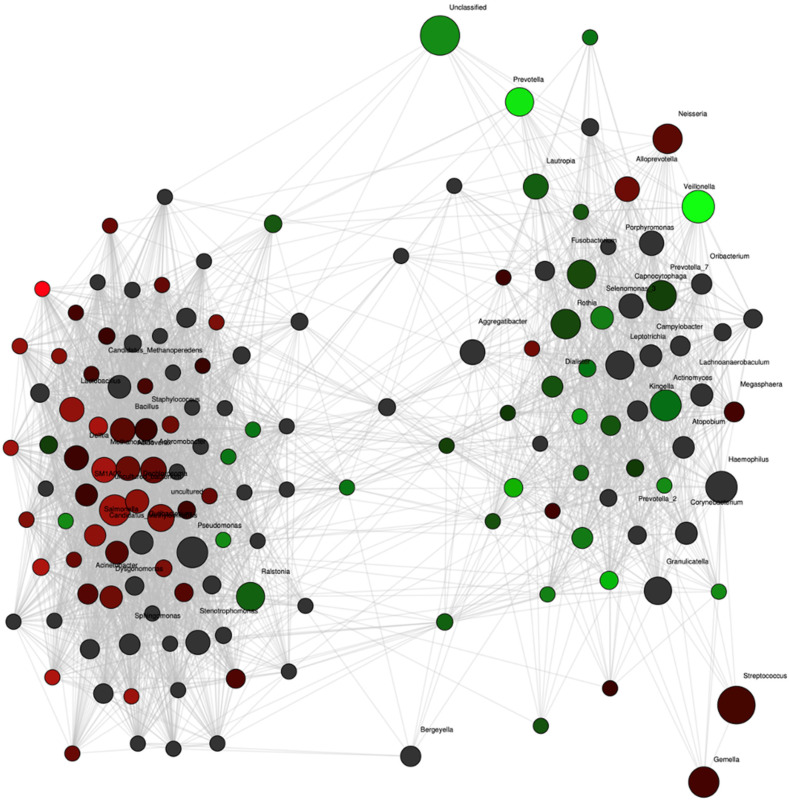
Network analysis at the genus level for pregnant women with preeclampsia (red nodes, *n* = 12) and control (green nodes, *n* = 24) groups at 36 weeks of pregnancy. Genera are depicted as nodes. The color intensity indicates the strength of the associations with group. Significant positive correlation coefficients were calculated and adjusted for multiple testing with 500-fold permutations.

**Figure 4 nutrients-14-01139-f004:**
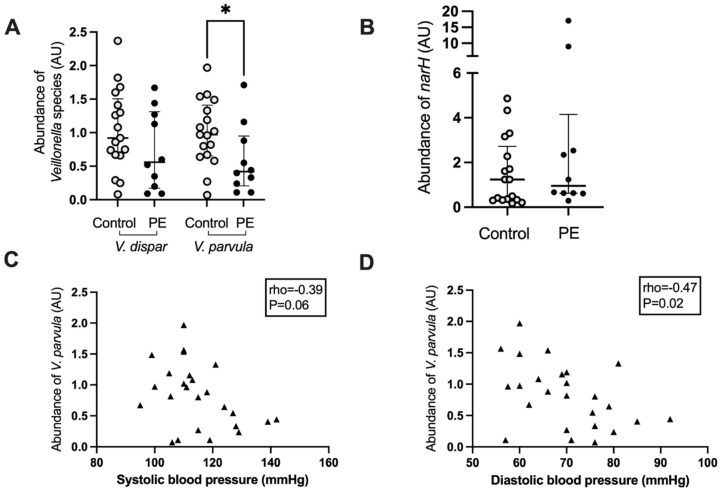
*Veillonella* species abundance (**A**) and nitrate reductase subunit narH (**B**) in the maternal oral microbiota of women who developed preeclampsia and controls. Correlation of maternal systolic (**C**) and diastolic (**D**) blood pressure with the abundance of narH in the oral microbiota. Light grey circles, samples from control women; black circles, samples from women who develop preeclampsia, black triangles, samples from all women; *, *p* < 0.05.

**Table 1 nutrients-14-01139-t001:** Clinical characteristics of participants at 28 weeks gestation.

Characteristic	Preeclampsia	Control	*p* Value
*n*	12	24	
Age (years)	32 (27.5–35.5)	32 (31.5–35.75)	0.66
BMI (kg/m^2^)	38.05 (35.68–43)	36.35 (34.50–41)	0.072
SBP (mmHg) *	128 (112–130) ^#^	110 (104–119)	0.006
DBP (mmHg) *	75 (67–83) ^#^	65 (60–73)	0.026
Glucose (mmol/L)	4.2 (4.1–4.5)	4.3 (4.0–4.5)	0.99
HbA1c (%)	5.4 (4.9–5.6)	5.1 (4.9–5.3)	0.23
C peptide (nmol/L)	0.6 (0.5–1.0)	0.5 (0.3–0.6)	0.068
Insulin (mU/L)	8.7 (4.5–10.0)	5.0 (3.4–7.1)	0.038
Cholesterol (mmol/L)	6.1 (5.6–7.0)	6.0 (5.5–6.7)	0.62
Triglyceride (mmol/L)	2.2 (1.8–2.9)	2.1 (1.9–2.9)	0.86
HDL (mmol/L)	1.7 (1.3–2.0)	1.6 (1.4–1.8)	0.5
LDL (mmol/L)	3.2 (2.6–4.5)	3.7 (3.1–4.1)	0.96
VLDL (mmol/L)	1.0 (0.9–1.4)	1.0 (0.8–1.3)	0.87
Pregnancy outcomes			
Infant birth weight	3613 (2915–3782)	3488 (3215–3704)	0.72
Infant gender (M/F)	7/5	11/13	0.48
Delivery mode Vaginal/Caesarean	5/7	13/11	0.47
Gestational age at delivery	39.4 (37.6–39.8)	39.7 (38.6–40.9)	0.15

Data are presented as the median with interquartile range. BMI: body mass index, SBP: systolic blood pressure, DBP: diastolic blood pressure, HbA1c: hemoglobin A1c, HDL: high-density lipoproteins, LDL: low-density lipoproteins, and VLDL: very-low-density lipoprotein. *, obtained at 36 weeks gestation; ^#^, 4 women were using anti-hypertensive medication, and 2 of these had chronic hypertension.

**Table 2 nutrients-14-01139-t002:** Significant correlations between oral bacteria and systolic or diastolic blood pressure.

Genus	Systolic Blood Pressure		Diastolic Blood Pressure	
	Rho	*p*-Value	Rho	*p*-Value
*Methanobrevibacter*	0.48	0.007	0.53	0.002
*Candidatus Brocadia*	0.48	0.007	-	-
*Uncultured organism*	0.48	0.007	0.34	0.044
*Veillonella*	−0.46	0.010	−0.38	0.036
*Desulfomicrobium*	0.45	0.013	-	-
*Mycoplasma*	0.44	0.014	0.38	0.037
*Actinophytocola*	0.42	0.019		
*Lachnospiraceae NK4A136*	0.41	0.025	0.47	0.008
*Roseomonas*	−0.39	0.031	-	-
*Aggregatibacter*	−0.38	0.037	-	-
*Dechloromonas*	0.37	0.042	-	-
*Prevotella*	−0.16	0.088	−0.14	0.091
*Actinophytocola*	-	-	0.49	0.005
*Peptococcus*	-	-	0.42	0.02
*Lysobacter*	-	-	0.39	0.035
*Ignavibacterium*	-	-	0.39	0.035

## Data Availability

Data will be made available upon reasonable request by contacting the corresponding author.

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
