# Peer review of "Reduced Abundance of Nitrate-Reducing Bacteria in the Oral Microbiota of Women with Future Preeclampsia"

_nutrients, 2022, doi:10.3390/nu14061139_

Round 1
Reviewer 1 Report
I commend the authors for undertaking their study "Reduced abundance of nitrate-reducing bacteria..." to evaluate whether the oral microbiota contribute to blood pressure regulation. While this concept has been explored in other studies, there are certainly gaps in knowledge on this topic.
Strengths:
- Well written manuscript with appropriate introductions
- Figures are well-designed and fairly easy to understand.
Weaknesses:
- This was a sub-study so there are limitations related to the original study design
- It is unclear if all women included in this study had received probiotics and what impact that might have had on the oral microbiota
- The sample size is very small. Additionally, mixing in two chronic hypertensive patients with superimposed preeclampsia makes the analysis even more difficult. When looking at your figures, it is unclear whether the underlying hypertensive disorder was accounted for.
- It also appears that there were more diabetics or insulin resistant women in the preeclamptic group which also must be accounted for
- In figure 4 B it looks like there may be two outliers in the preeclamptic group. Was an outlier statistical evaluation undertaken? Are these the women with chronic hypertension?
- The conclusion is slightly overstated. I'd be less certain in your conclusion with a study this small.
Author Response
I commend the authors for undertaking their study "Reduced abundance of nitrate-reducing bacteria..." to evaluate whether the oral microbiota contribute to blood pressure regulation. While this concept has been explored in other studies, there are certainly gaps in knowledge on this topic.
Strengths:
- Well written manuscript with appropriate introductions
- Figures are well-designed and fairly easy to understand.
Thank you for your kind comments.
Weaknesses:
- This was a sub-study so there are limitations related to the original study design
The reviewer is of course correct here. The fact that this analysis was done as a sub-study of a larger RCT does bring with it limitations that we have tried to address or acknowledge in the limitations paragraph.
- It is unclear if all women included in this study had received probiotics and what impact that might have had on the oral microbiota
Thank you, this is a very important point and our apologies for not addressing this in the original submission. The probiotics were taken by just over half of the study cohort: they were capsules designed to remain intact in the acidic environment of the stomach. It is therefore highly unlikely that the participants that took probiotics would have been exposed to any bacteria in the oral cavity. However, we did check whether there were any differences in the abundances of bacteria between the groups and there were none. We have now added that information to the Discussion section, the sentence reads:
“The sample size is relatively small and included women who were assigned probiotics. However, there were no significant differences in bacterial abundances between the women receiving probiotics or placebo capsules, which were designed to withstand the acidic content of the stomach and therefore unlikely to directly affect the composition of the oral microbiota.”
- The sample size is very small. Additionally, mixing in two chronic hypertensive patients with superimposed preeclampsia makes the analysis even more difficult. When looking at your figures, it is unclear whether the underlying hypertensive disorder was accounted for.
- It also appears that there were more diabetics or insulin resistant women in the preeclamptic group which also must be accounted for
Thank you for pointing this out. There were only two women who had chronic hypertension among our women with preeclampsia and three of the women with preeclampsia developed GDM. We did a sensitivity analysis and this did not change the results. We have clarified this in the revised manuscript in the methods section, statistical analysis:
“Sensitivity analyses were performed excluding the two women with chronic hypertension and separately for the three women who developed GDM but this did not alter the results.”
- In figure 4 B it looks like there may be two outliers in the preeclamptic group. Was an outlier statistical evaluation undertaken? Are these the women with chronic hypertension?
Thank you for this comment. We performed non-parametric analysis in this study which reduces the impact of large variability in the results given that it is based on rank testing rather than using the actual values for the testing. We critically evaluated both the technical aspects of the analysis, where we did not find any reason to exclude these samples. In addition, we assessed if the high expression of narH was related to any clinical characteristic of these women but this was not the case. One of the samples with high narH expression indeed came from a participant with chronic hypertension but the other participant with chronic hypertension had narH expression below the median for the preeclampsia group. The other sample with high narH expression in the preeclampsia group did not have chronic hypertension or GDM. Given that we cannot find a technically or clinically valid reason for excluding these samples, we have elected to show the data as is.
- The conclusion is slightly overstated. I'd be less certain in your conclusion with a study this small.
Thank you, we have toned down the conclusion. It now reads:
“In summary, this study indicates that the abundances of nitrate-reducing bacteria in the oral microbiota are reduced in pregnant women with future late-onset preeclampsia suggesting a potential pathway for prevention.”
Reviewer 2 Report
This manuscript studies the relationship between blood pressure and the diversity of the oral microbiota in normal pregnant patients and those with preeclampsia. The experimental design is adequate and the results obtained, although they have limitations, are of some interest. The main concern of this reviewer is that the manuscript is within the scope of the journal. The connection with diet or eating disorders is week. Thus, the significance of the content and the interest for the readers of the journal is very limited.
Minor points
- The bibliography is quite scarce. Authors should consider including recent publications such as Jang H et al, Sci Rep (2021) and Willmott T et al, Front Cell Infect Microbio (2020).
- 2. Some of the figures and result texts are confused (example, Fig 2 and explanation of Fig 3).
Author Response
This manuscript studies the relationship between blood pressure and the diversity of the oral microbiota in normal pregnant patients and those with preeclampsia. The experimental design is adequate and the results obtained, although they have limitations, are of some interest. The main concern of this reviewer is that the manuscript is within the scope of the journal. The connection with diet or eating disorders is week. Thus, the significance of the content and the interest for the readers of the journal is very limited.
Thank you. We respectfully disagree with the statement that there is a weak link between diet and our study. There is a direct link between the intake of nitrate-containing food items, mainly leafy green vegetables, and the oral production of NO through the oral microbiota. Our study shows that there is a difference in nitrate-reducing bacteria which could provide a target for intervention either by increasing their abundance through supplementation with the substrates on which they grow or through supplementation with other bacteria.
Minor points
- The bibliography is quite scarce. Authors should consider including recent publications such as Jang H et al, Sci Rep (2021) and Willmott T et al, Front Cell Infect Microbio (2020).
Thank you, we have added these and two more references to the background. This new section now reads:
“The oral microbiome in pregnancy has been reported to be altered from the non-pregnant state but remains stable in its overall diversity and composition during pregnancy [7]. Studies using culturing techniques on gingival plaques in women with preeclampsia have shown increases in the rates of periodontitis and it associated pathogens, Porphyromonas gingivalis, Tenerella forsynthensis and Eikenella corrodens [14,15]. However, there have been no studies using current sequencing techniques to evaluate the composition of the oral microbiome in women who develop preeclampsia.”
- 2. Some of the figures and result texts are confused (example, Fig 2 and explanation of Fig 3).
Thank you, this has been checked.
Round 2
Reviewer 1 Report
Nice job answering the critiques I made and strengthening your manuscript. I think the revised form is more clear.